# Methods for Lowering the Power Consumption of OS-Based Adaptive Deep Brain Stimulation Controllers

**DOI:** 10.3390/s21072349

**Published:** 2021-03-28

**Authors:** Roberto Rodriguez-Zurrunero, Alvaro Araujo, Madeleine M. Lowery

**Affiliations:** 1B105 Electronic Systems Lab. ETSI Telecomunicación, Universidad Politécnica de Madrid, 28040 Madrid, Spain; araujo@b105.upm.es; 2School of Electrical, Electronical and Communications Engineering, University College Dublin, Belfield, Dublin 4, Ireland; madeleine.lowery@ucd.ie

**Keywords:** DBS, adaptive DBS, operating system, embedded system, microcontroller, Parkinson dDisease, neuromodulation, electrical stimulation

## Abstract

The identification of a new generation of adaptive strategies for deep brain stimulation (DBS) will require the development of mixed hardware–software systems for testing and implementing such controllers clinically. Towards this aim, introducing an operating system (OS) that provides high-level features (multitasking, hardware abstraction, and dynamic operation) as the core element of adaptive deep brain stimulation (aDBS) controllers could expand the capabilities and development speed of new control strategies. However, such software frameworks also introduce substantial power consumption overhead that could render this solution unfeasible for implantable devices. To address this, in this work four techniques to reduce this overhead are proposed and evaluated: a tick-less idle operation mode, reduced and dynamic sampling, buffered read mode, and duty cycling. A dual threshold adaptive deep brain stimulation algorithm for suppressing pathological oscillatory neural activity was implemented along with the proposed energy saving techniques on an energy-efficient OS, YetiOS, running on a STM32L476RE microcontroller. The system was then tested using an emulation environment coupled to a mean field model of the parkinsonian basal ganglia to simulate local field potential (LFPs) which acted as a biomarker for the controller. The OS-based controller alone introduced a power consumption overhead of 10.03 mW for a sampling rate of 1 kHz. This was reduced to 12 μW by applying the proposed tick-less idle mode, dynamic sampling, buffered read and duty cycling techniques. The OS-based controller using the proposed methods can facilitate rapid and flexible testing and implementation of new control methods. Furthermore, the approach has the potential to become a central element in future implantable devices to enable energy-efficient implementation of a wide range of control algorithms across different neurological conditions and hardware platforms.

## 1. Introduction

Over the past two decades, Deep Brain Stimulation (DBS) has become established as an effective surgical therapy to reduce the symptoms of several neurological conditions including Parkinson’s Disease (PD), essential tremor, dystonia, epilepsy, and severe obsessive-compulsive disorder [1,2,3]. More recently, a gradual increase in understanding of the mechanisms by which DBS exerts its therapeutic effects has led to the emergence of closed-loop neuromodulation or adaptive DBS (aDBS) techniques [4,5,6,7]. Using this approach, instead of delivering a constant stimulation signal, the stimulation parameters are adjusted in response to biomarkers indicative of patient symptoms [8,9,10,11]. These new techniques present potential benefits both in the effectiveness of the therapies, by delivering the stimulation required to control symptoms while minimizing stimulation-induced side effects, and in reducing power consumption of the stimulation devices [5,12,13]. There is thus currently considerable interest among clinicians, scientists, and device manufacturers in developing new aDBS algorithms and devices to fully exploit the benefits of this approach [14,15,16].

The electronic devices required to deliver aDBS will be more complex than traditional stimulation devices since they will require specialized hardware and software to detect, record and analyze the biomarker data [17,18], in addition to a controller module capable of changing the stimulation parameters in response to the detected biomarker [19]. A typical such device will comprise acquisition, stimulation, controller, wireless communications, and power management modules. As energy efficiency is one of the most critical constraints of these types of implantable devices, research in this field has focused on designing custom-developed hardware devices to achieve the best performance with the lowest power consumption possible [20,21,22]. Rechargeable systems have also been developed to address this important constraint as in the Activa RC device from Medtronic [23]

New device designs usually include a processor unit as the core element of the controller module enabling firmware upgrades and enhancing the flexibility to implement new adaptive algorithms. A prototype of an aDBS device using a custom Cortex-M3 processor to provide some programming flexibility was presented by Cong et al. [24]. This prototype design was the basis for the most recent commercial device available, the Summit RC+S from Medtronic [19], which includes a 68HC11 processor unit providing firmware update with reported power consumption as low as 2.5 mW. Other processor-based devices have also been developed providing improved programming flexibility depending on the technology they are using, e.g., an open-source Arduino platform [25], a MSP430 low-power microcontroller [26], a SmartFusion 2 SoC [27] or a low power Flash-FPGA [28].

While firmware development is gaining importance, a common flexible framework could be very useful in speeding up the development time of new adaptive DBS techniques, reducing the time to deployment of new enhanced techniques in devices in PD patients to improve quality of life. However, as far as we know, the inclusion of more complex programming architectures in these devices has not yet been analyzed in depth. Specifically, none of these devices has implemented an embedded OS [29] as the core element of the controller module. An embedded OS has the potential to greatly expand the capability of closed-loop neuromodulation devices by providing a firmware architecture that can enhance software development speed by providing high-level features and services to the developers [30]. It provides a hardware abstraction layer, multitasking management, and dynamic resource management, so advanced applications can be easily developed by developers that are not hardware experts (reducing the required hardware background of algorithms researchers). In addition, an OS provides a scalable and upgradeable common framework to develop applications for different hardware platforms and for different application fields, i.e., for the treatment of different neurological conditions. These advanced features imply an extra complexity in the low-level firmware which results in an overhead in terms of memory usage, processing time, and power consumption [31,32,33], which may not be affordable in implantable devices. Thus, it is crucial to implement power management techniques in these OSs to achieve sufficient energy efficiency to make them feasible for neuromodulation systems such as aDBS.

To address this need, in this work we present four different techniques to reduce the power consumption in a device running an aDBS algorithm on top of an OS. In the OS kernel, automatic “tick-less” operation is implemented (main internal clocks shutdown) to reduce power consumption [34]. To further reduce power consumption overhead from a driver level perspective, a buffered read mode is introduced in combination with reduced and dynamic sampling. On top of the OS, a duty cycling mechanism is also proposed, which saves energy by periodically disabling the running aDBS algorithm.

In order to evaluate the proposed techniques, an aDBS algorithm for Parkinson’s disease that modulates stimulation amplitude based on beta band (15–30 Hz) neural activity—shown to be correlated with symptoms of Parkinson’s disease—was implemented on YetiOS [35]. A real-time emulation environment for PC based on a mathematical model of the parkinsonian basal ganglia was developed to evaluate the proposed solution. The experimental evaluation for different operation modes and sample rates, demonstrates the feasibility of an OS-based aDBS device in terms of performance and power consumption (as low as 12 μW) while taking advantage of the high-level features provided by the OS. This work represents a first step towards introducing more advanced software architectures to improve the capabilities and flexibility of neuromodulation devices. Although the OS-based approach is initially focused on reducing the development time of new aDBS techniques in research environments, sharing a common software framework with implanted devices could greatly reduce the time from when an algorithm passes clinical trials until it can be used in PD patients.

## 2. Materials and Methods

To experimentally evaluate the proposed techniques, a dual threshold aDBS algorithm was implemented in a general-purpose STM32L476RE microcontroller—ARM Cortex-M4 core—running the YetiOS operating system. YetiOS is an open-sourced energy efficient OS designed for resource-constrained devices which provides an adaptive engine to easily implement adaptive algorithms. A real-time emulation environment has been used for the acquisition and stimulation modules. This enables the overhead in terms of power consumption introduced by the proposed OS-based controller module to be measured directly. The power consumption of the other modules (acquisition, stimulation, and power management) is not considered here.

### 2.1. OS for sDBS with Tick-Less Mode and Buffered Sampling

YetiOS, based on the well-known operating system FreeRTOS [36], was implemented in the STM32L476RE microcontroller. A low power manager has been implemented in YetiOS which automatically sets the microcontroller in tick-less mode when no tasks need to be executed. Unlike other tick-less implementations, such as that used in FreeRTOS, where the main system timer needs to be always active (preventing the microcontroller entering the lowest power consumption modes), the tick-less mode implemented shutdowns the main system clocks, as well as most peripherals, keeping only a low frequency low power timer running in order to reach the lowest power consumption possible. Multitasking capabilities of YetiOS enables implementation of the aDBS algorithm using two concurrent tasks, which allows isolation of the sampling and processing programming. In this way, it is possible to implement different sampling techniques, such as single sample read and buffered read modes, while keeping the same processing algorithm. When using the single sample read mode, Figure 1, the sampling task (green) reads a single sample from the serial peripheral interface (SPI) each time a sample is available in the analog to digital converter (ADC) and stores it in local memory. When enough samples are stored, the processing task (blue) is launched to perform the operations determined by the aDBS algorithm to set the stimulation amplitude. Since both tasks run concurrently and the sampling task must not be delayed, it has higher priority than the processing task. If new samples are available in the ADC, the sampling task thus reads them, interrupting the processing task if it is running as shown in the grey regions in Figure 1. In addition, this figure shows in red regions where the tick-less mode is entered when no tasks are executed. Finally, it is important to note that entering and exiting a task in the OS always requires instructions to be executed, implying an execution overhead, shown in brown, which translates to power consumption overhead. The higher the sample rate or the number of context changes, the greater the overhead introduced by the operating system. It is important to highlight that the overhead introduced by non-ideal switching results in a maximum sample frequency in which using tick-less mode saves energy. For example, if the switching overhead time (brown) for each sample is 50 μs, we would have a maximum sample frequency of 20 KHz. However, this maximum frequency is even lower due to the processing time required by the sampling and processing tasks.

To reduce this execution overhead, a buffered read operation mode was implemented. An example of the buffered read mode is presented in Figure 2, in which samples are read in blocks of 3 samples by the sampling task. The execution overhead can be thus reduced, so the microcontroller remains in low power tick-less mode for a longer time than when reading each sample individually. Power consumption was measured when using buffer sizes of 8, 16, and 32 samples.

The buffered read mode may introduce a latency if the time required to acquire all the samples of the buffer is larger than the period in which they are processed. Therefore, for each sample rate a maximum buffer size that does not introduce any extra latency is defined.

### 2.2. Sample Rate Management and Duty Cycling Mode

Sample rate (*sr*) has an important impact on power consumption due to the execution overhead introduced by entering and exiting the sampling task. The sample rate should be as low as possible to reduce power consumption, however reducing the sample rate could have a negative impact in the performance of the control algorithm. A lower sample rate usually implies higher aliasing noise due to the non-ideal nature of the analog filters, hence the signal to noise ratio benefits of oversampling may not be exploited. It is well established that sampling with a rate *N* times higher than the Nyquist rate results in a signal to noise ratio (SNR) improvement of $\sqrt{N}$. Thus, reducing the sampling rate could impact on 
the decisions made by the adaptive algorithm. The OS controller enables the 
sample rate to be configured and changed dynamically without modifying the aDBS 
algorithm. 

Sample LFP data recorded from a patient with PD at a range of DBS amplitudes was examined for different sampling rates, patient 1 in the work of Davidson et al. [37]. The data were recorded at the Department of Clinical Neurology, University of Oxford, using a single channel with a pass band of 4–40 Hz and an original sample rate of 2.2 kHz. The recorded signal was decimated to 1 kHz and 100 Hz and the aDBS algorithm was executed to examine how sampling rate can affect the output decision of the control algorithm using physiological LFP signals. The stimulation product (SP) is introduced to quantify the performance of the aDBS algorithm for different sample rates. This parameter represents the product of the root mean square (RMS) value of the beta band signal ($\beta_{RMS}$) and the RMS value of the stimulation amplitude ($A_{RMS}$), Equation (1). The lower the SP, the better the aDBS performance is deemed to be, since the target is to reduce beta band amplitude while minimizing stimulation amplitude.
(1)$$SP=\beta_{RMS}\cdot A_{RMS}$$

The SP was used to evaluate the performance when running the real-time emulator to generate different stimulation signals which causes variable beta band signal readings.

To overcome degradation in performance at lower sampling rates, while minimizing the associated power consumption, a dynamic sample rate algorithm was implemented using the dynamic capabilities enabled by the OS-based controller. With dynamic sampling, the decisions made using different sample rates are periodically evaluated to raise or reduce the sampling rate when the number of mismatches in those decisions reach a threshold value. In this way, intermediate power consumptions values between different sampling rates are achieved while maintaining the controller performance.

Finally, a duty cycling mechanism was implemented on top of the OS to further reduce the power consumption. This simple technique is implemented on a periodic independent task that enables the sampling and processing tasks for a short time $t_{on}$ during a large period $T_{s}$, and disables them the rest of the time. Therefore, the aDBS algorithm runs only when these tasks are active, with stimulation amplitude remaining constant the rest of the time. A fixed $t_{on}$ of 5 s is used, while the period $T_{s}$ is varied to evaluate the power consumption for different duty cycle (DC) values.

### 2.3. Dual Threshold Adaptive Deep Brain Stimulation (aDBS) Algorithm

To demonstrate and evaluate the proposed system, an adaptive DBS algorithm based on the recent work of Velisar et al. [16] was implemented in the processing task. The algorithm adapts the stimulation amplitude based on the subthalamic nucleus (STN) LFP beta band activity recorded at non-stimulating contacts on the DBS electrode. A dual threshold policy is used whereby beta power is maintained between two defined thresholds by increasing or decreasing the stimulation amplitude when the beta power lies or above or below the upper and lower thresholds, respectively. An increase in both efficacy and efficiency was demonstrated in PD patients using this approach [16].

The complete adaptive stimulation algorithm implemented here is presented in Figure 3. The controller continuously acquires the simulated 16-bit sampled LFP data and stores them in local memory to fill a moving window of size *N*. Each window overlaps *k* samples with the previous one, so the adaptive algorithm is executed each time *N-k* samples are obtained. The implementation of the adaptive algorithm is described as follows:

First, each one of the last N-k samples is converted to a voltage value (range 0–1.2 V) represented by a single precision 32-bit floating point number.A bandpass finite impulse response (FIR) filter is applied for these N-k samples. The FIR filter has M number of coefficients. This filter was designed to isolate the beta band oscillations, with pass band from 10 Hz to 30 Hz, and stop frequencies at 5 Hz and 35 Hz. The coefficients of the filter for each sample rate were obtained using Matlab filter design tools and setting a −3 dB gain in the limits of the pass band (10–30 Hz), and at least −10 dB gain beyond the cut-off frequencies (below 5 Hz and above 35 Hz). The functions provided by the ARM CMSIS DSP library [38] were used to implement the FIR filter in the STM32L476RE microcontroller.The arithmetic mean of the filtered N-k samples is then calculated and subtracted from each sample to remove any residual DC component within the signal.The LFP signal energy E from the last N samples is calculated as in Equation (2), where s[n] represents the filtered signal samples. This signal energy was chosen as the feature used to perform the dual threshold algorithm, since it requires only one multiplication operation and one sum operation for each sample in the window.
(2)$$E=\;\overset{N-1}{\underset{n=0}{\sum }}{\left|s\left[n\right]\right|}^{2}$$Finally, the obtained LFP signal energy is evaluated to set the stimulation amplitude, as in Equation (3). If the energy is larger than the higher threshold, $TH_{1}$, the stimulation amplitude *A* is increased one amplitude step $A_{STEP}$, while it is decreased one amplitude step if the energy is below the lower threshold, $TH_{2}$. The stimulation amplitude remains constant if the energy remains between the two threshold values. The stimulation amplitude step $A_{STEP}$ is set to 0.1 a.u. providing high granularity, similar to the approach used by Velisar et al. [16] where a 0.1 V step size was used.
(3)$$A{\left(E\right)}_{i+1}=\;\{\begin{array}{c} A_{i}+\;A_{STEP}\;\;\;\;if\;\;\;E>\;TH_{1}\\ A_{i}-\;A_{STEP}\;\;\;\;if\;\;\;E<\;TH_{2}\\ A_{i}\;\;\;\;\;\;if\;\;\;\;E\;\in \;\left[\;TH_{1},\;\;TH_{2}\right] \end{array}$$

The configured thresholds were set and calibrated after performing empirical tests in the emulation environment to obtain an appropriate adaptive stimulation behaviour similar to that reported clinically [16]. The parameters for the adaptive algorithm are summarized in the Table 1. The decision time is 256 ms for all the sample rates except for 100 Hz rate in which it is 320 ms, since it depends on the non-overlapping samples number *N-k*. These decision times led to the maximum buffer size of 32 samples when using the buffered read mode to avoid introducing additional latency when sampling at 100 Hz. This maximum buffer size increases to 64, 128, and 256 samples for sample rates of 250 Hz, 500 Hz and 1000 Hz, respectively. Finally, it is important to mention that the aDBS algorithm works in the same way for the different sample rates and read modes used in the experiments.

### 2.4. Real Time Emulation Environment for Adaptive DBS

A real time emulation environment for PC, Figure 4, was developed to evaluate the OS-based controller solution. The core of the real time emulator, Figure 5, is based on a mean-field model of the basal ganglia GPe-STN network. Mean-field models [39], such as that used here, provide a low-dimensional representation of the dynamics of large populations of synchronous neurons to reduce the complexity of the system under investigation while retaining key behaviors of the networks involved.

This model simulates the generation of pathological beta-band neural oscillations in Parkinson disease and their response to STN stimulation [40]. It has previously been shown that the suppression of pathological beta band activity in the local field potential (LFP) during DBS in Parkinsonian patients with implanted DBS is well-described by the model [37], Equations (4) and (5). The model was designed using Mathworks Simulink software, and then exported to C++ code, Figure 5, where:(4)$$G\left(s\right)=\;\frac{k}{{\left(s+b\right)}^{2}}$$
(5)$$NL,\;h:\;y_{1}=\;\frac{2}{\pi }arctan\frac{x_{1}}{h};\;NL,\;g:\;y_{2}=\;\frac{2}{\pi }arctan\frac{x_{2}}{g}$$

The GPe and STN are modelled as nonlinear sigmoidal algebraic elements *NL*, where *h* and *g* define the steepness of the characteristic, followed by linear blocks defined by their transfer functions in the Laplace domain *G*(*s*), where *k* and *b* are constants. *x*_1_ and *x*_2_ represent the mean field deviation from zero of the LFP of the population of synchronous neurons in the GPe and STN, respectively. *y*_1_ and *y*_2_ are the outputs of the sigmoidal arctan function, following the approach of classic mean field models, to represent the relationship between the average membrane depolarization and the average firing rate of the neural population. Oscillations representing beta band activity of the parkinsonian local field potential emerge within the network as the strength of coupling between the STN and GPe increases, *h* or *g* decrease. The parameters were set to simulate oscillatory activity in the GPe-STN loop in the beta band, specifically, at 22.5 Hz. Following the approach presented by de Paor et al. [40], oscillatory activity within the network at 22.5 Hz was obtained by setting $k=b^{2}=19985$, g=1, and h=rand[0.100, 0.101]. The h parameter simulates the increase in strength of synaptic coupling within the network in response to dopamine depletion in Parkinson’s disease and was assigned a bounded uniform random value that changed every 5 s to simulate the amplitude variations in the oscillatory activity. An additive noise source (normal distribution random generator, $\mu =0,\;\sigma^{2}=0.001$ arbitrary units (a.u.)), was included to simulate the input noise at the electrodes, while a 250 Hz low pass filter represented the acquisition filters present in the device.

The emulator runs on a Windows PC and provides a graphical interface to configure the model parameters, display and record the LFP and stimulation signals and manage the aDBS controller. The emulator uses a USB interface for streaming the simulated LFP data and configuring the stimulator and is connected to an USB to SPI bridge device that emulates a 16-bit ADC and a configurable stimulator, Figure 4. This bridge device transmits the simulated raw LFP data using a SPI interface with a configurable sample rate of up to 1000 Hz to the controller device which is an STM32L476RE microcontroller running the aDBS algorithm in YetiOS. The full-duplex SPI interface also allows the controller to configure the stimulation parameters in the emulator. Specifically, the stimulation pulses amplitude *A* a.u. is configured by the aDBS algorithm, while the pulses frequency and width are set to 140 Hz and 60 μs, respectively. The real time emulator is available at [41].

### 2.5. Experimental Conditions

Each test was performed within the emulation environment for 250 s to measure the power consumption of the test device executing the dual threshold aDBS algorithm in YetiOS. The power consumption was measured using a Keysight B2901A precision source measurement unit supplying the device at 3.3 V and with 2.5 ms acquisition time, so 100,000 power consumption samples are acquired for each test (250 s). Power consumption data as well as beta band and stimulation amplitude data of the tests performed are accessible on a public dataset [42].

The code was compiled using GCC with level 3 optimization (O3). It required 58 Kb of FLASH non-volatile memory and 25 Kb of RAM memory which is much lower than the totally available in the STM32L476RE microcontroller: 512 Kb of Flash and 128 Kb of RAM. The processing time required for the aDBS algorithm to execute was different for each evaluated sample rate-since the window samples to be processed and the FIR filter taps number are different-being 2103 µs for 1000 Hz, 636 µs for 500 Hz, 267 µs for Hz, and 124 µs for 100 Hz.

## 3. Results

In order to set the threshold values $TH_{1}$ and $TH_{2}$ of the dual threshold aDBS algorithm the *SP* has been measured, Figure 6. From the experiments performed we obtained the lowest *SP* values (best performance) for $TH_{1}=0.008\;a.u.$ and $TH_{2}=0.0004\;a.u.$ These values are used as the default configuration of the dual threshold algorithm in all the tests performed. 

An example of raw LFP data for a PD patient [37] for different stimulation amplitudes is presented in Figure 7a, along with the calculated signal energy Figure 7b, for the maximum and the minimum allowed sample rates. The red line represents the amplitude of the stimulation signal applied in the clinical experiment, while the green lines represent arbitrary threshold values that may be used by a dual threshold algorithm to make the stimulation decisions. Although the overall signal energy appears similar for both sample rates, as shown in Figure 7c, at certain points, a different decision could be made when evaluating the signal energy with the different sample rates due to degradation of the SNR for reduced sample rates. It is not yet clear whether such differences or resulting delays in the decisions made for different sample rates correspond to a degradation in clinical performance. This is likely to be algorithm dependent, though it is feasible that resulting delays of up to 1 s in making the correct decision may have an adverse effect on algorithm performance.

Similar to Figure 7, Figure 8 represents the raw LFP data as well as the signal energy in an experiment using the emulation environment with the stimulation amplitude progressively increased. The oscillatory activity is reduced when applying high stimulation amplitude, similar to that observed in the experimental LFP data. The behavior of the dual threshold aDBS algorithm for the different sample rates, and with the duty cycling mode are presented in Figure 9. Differences between the applied stimulation signals are observed when acquiring with 1000 Hz and at lower sample rates. In the duty cycling mode, the aDBS algorithm is dormant most of the time, so the changes in the stimulation signal happens less frequently. As it can be observed, there are differences in both the stimulation signal and the LFP readings when changing the sample rate, which would translate in different algorithm performances in each case.

The measured power consumption of the controller for different sample rates is presented in Figure 10. The power consumption is compared for the single and buffered sample read modes. Introducing the buffered read mode (32 samples buffer size) enables the power consumption to be greatly reduced, reaching as low as 50 μW for 100 Hz, as compared with 400 μW for the single sample read mode. Using 8 and 16 samples in the read buffer provides intermediate power consumption values. Sample rate also had a large impact on the power consumption of the controller as predicted, but resulted in a performance degradation in terms of *SP*. In the duty cycling mode, the power consumption was lowest, but the performance, in terms of *SP*, was also the lowest (highest *SP*) since the acquisition was disabled most of the time. 

The power consumption for different duty cycling configurations is presented in Figure 11, reaching as low as 12 μW for the lowest sample rate (100 Hz) and duty cycle evaluated (5%). The power consumption of the dynamic sampling rate example application was also evaluated reaching 763 μW for single sample read mode and 102 μW for 32 samples buffered read mode. Thus, by using this simple dynamic application, the power consumption measured is a midpoint value between the one obtained for the lowest sample rate and higher sample rates.

## 4. Discussion

Using an OS, the programming of aDBS algorithms can be organized in different concurrent tasks, and algorithms can be programmed without expert knowledge of the hardware components. In addition, dynamic switching between stimulation control algorithms, for example, to target different disease symptoms or depending on the patient’s activity, is easily implemented. The OS provides upgradeable capabilities, so alternative adaptive stimulation techniques can be implemented and upgraded without changing hardware. Conversely, as the OS isolates the algorithms implementation from the hardware components, the code developed on an OS for a specific algorithm would be the same if a different HW device is used. The dynamic capabilities of the OS-based solution allow techniques to optimize aDBS performance to be implemented, trading-off between stimulation performance and power consumption, as with the proposed example of the dynamic sample rate mode. Finally, the OS provides a framework to develop algorithms to detect patient state based on different biomarkers, e.g., EEG, EMG, and accelerometry, to automatically switch to duty cycling or alternative modes. A specific control algorithm (dual threshold) was implemented as an example of a recently proposed approach that has demonstrated to be clinically effective [16]. However, other aDBS algorithms can be similarly implemented using the OS-based controller.

The main drawback of the proposed system is the overhead in terms of power consumption introduced by the OS-based controller to the complete aDBS device. The measured power consumption of the OS-based controller reached up to 10.027 mW for the maximum evaluated sample rate, without implementing any technique to reduce power consumption. Using the automatic tick-idle operation with all the main clocks shutdown, the power consumption was considerably reduced to 3.904 mW. Further reductions in power consumption are possible by lowering the sample rate or using a buffered ADC. However, it is important to note that lowering the sample rate could produce a performance degradation (reduced SNR and increased SP), Figure 10, and the buffered read mode requires the ADCs implemented in aDBS devices to provide local sample buffers. The duty cycling mode proposed here also allows to large reductions in power consumption, at the expense of some performance degradation. A power consumption as low as 50 μW is achieved at 100 Hz using the 32 samples buffered read mode and can be further reduced to 12 μW with 5% duty cycle. These power consumption values would be affordable even with commercial implantable devices such as the Medtronic Summit RC+S [19]. The reported power consumption of the Summit RC+S device is 2.5 mW, so the additional power consumption introduced by the OS-based controller would be almost negligible in this case. For other research devices such as the high-end WAND [27], the impact is even lower since it consumes up to 172 mW. As that device includes an ARM Cortex-M3 processor unit similar to that used here, OS support to the device can be provided with limited development effort. Furthermore, rechargeable batteries used in many new DBS devices, reduce power consumption constraints, rendering the overhead introduced by the OS acceptable.

To test the performance effects of lowering the sample rate and using the duty cycling mode we have defined the parameter *SP*. However, further studies with clinically acquired data should be done in order to verify the actual clinical performance and to calibrate the different threshold values. Although a performance degradation in terms of *SP* is observed when reducing the sample rate and duty cycle, it is possible that this does not translate to a reduction in clinical performance in terms of control of patient symptoms and that acquisition with lower sample rates is sufficient. Further experimental and clinical investigations are required to address these types of questions and establish the sensitivity of specific aDBS algorithms and implementation details in patients. These types of studies are facilitated by the fast-development features provided by the OS. Finally, in addition to implementation constraints addressed by the introduction of the OS-based controller, the introduction of any new control system is governed by regulatory and safety issues. Prior to implementation in patients, these must be addressed, and appropriate risk mitigation strategies implemented considering technical standards for medical device hardware and software (e.g., IEC 60601- 1–10, and IEC 62304, ISO 14971). This can be achieved through the application of appropriate design frameworks such as that proposed by Gunduz et al. [43]. Finally, it is important to note that the techniques proposed in this paper could also be used in other clinical devices to implement an OS-based controller. For example, a similar approach may be used in other next generation intracorporeal implants [44] and also other closed loop neuromodulation devices such as those for epilepsy symptoms treatment or vagal nerve stimulation. 

## 5. Conclusions

An OS-based controller provides a wide range of programming flexibility for the development of aDBS and neuromodulation techniques that could enable advances in this field to be rapidly implemented. In this work, potential overhead in terms of additional power consumption introduced by the OS is mitigated through the implementation of an automatic tick-less idle mode, reduced sample rate, buffered read mode and duty cycling mode. A dynamic sample rate is presented to demonstrate the dynamic capabilities provided by the OS. The additional power consumption introduced by the OS applying the proposed techniques is thereby reduced to negligible levels (down to 12 μW) when compared with power consumption of currently available devices. The proposed OS approach has the potential to expand aDBS capability and development speed with an affordable cost in terms of power consumption. Although developed as a prototype for research purposes, it has the potential to become a key element in future commercial devices. However, as a higher abstraction level is provided by the OS, new design issues, such as security, software risk management, and real-time and wireless protocols programming, are introduced and will need to be addressed.

## Figures and Tables

**Figure 1 sensors-21-02349-f001:**
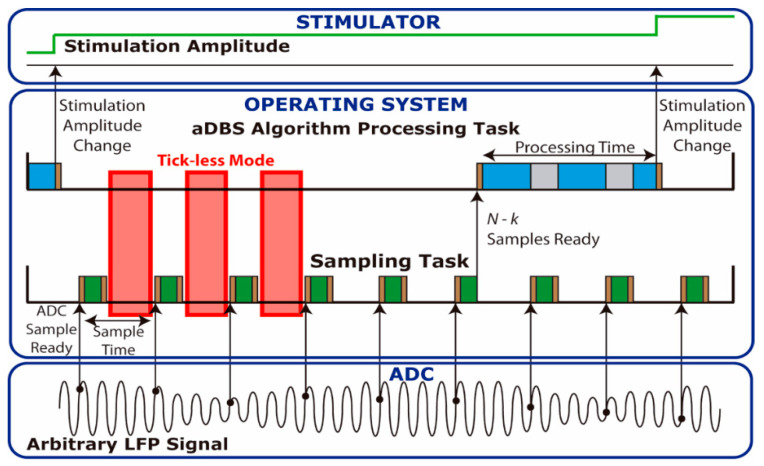
Tasks executed in the operating system (OS) to implement the adaptive deep brain stimulation (aDBS) algorithm reading from the analog to digital converter (ADC) each sample one by one. On the bottom, the original local field potential (LFP) signal captured by the ADC is represented. At each sample time, the sampling task reads a sample (green) and stores it for later processing. Once enough samples *N-k* needed for the dual threshold algorithm are ready, the processing task runs the aDBS algorithm (blue). When the processing is complete, a decision is made to change the stimulation amplitude. If there are no tasks running, the processor enters tick-less mode (red). There are always some overhead when switching between tasks and when going to and from tick-less mode (brown).

**Figure 2 sensors-21-02349-f002:**
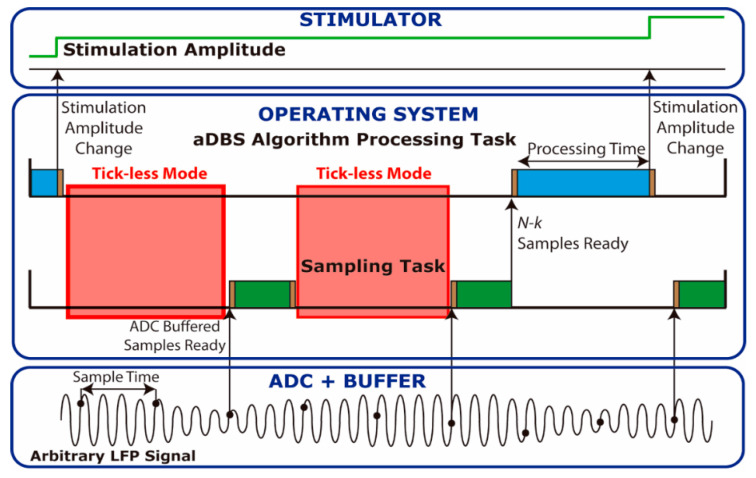
Tasks executed in the operating system (OS) to implement the adaptive deep brain stimulation (aDBS) algorithm using the buffered read mode. In this example, the samples are read in blocks of 3 from the analog to digital converter (ADC). On the bottom, the original local field potential (LFP) signal captured by the ADC is represented. When the read buffer is full (3 samples in this case), the sampling task reads a group of samples (green) and stores them for later processing. Once enough samples *N-k* needed for the dual threshold algorithm are ready, the processing task runs the aDBS algorithm (blue). When the processing is done, a decision is made to change the stimulation amplitude. If there are no tasks running, the processor enters tick-less mode (red). There are always some overhead when switching between tasks and when going to and from tick-less mode (brown).

**Figure 3 sensors-21-02349-f003:**
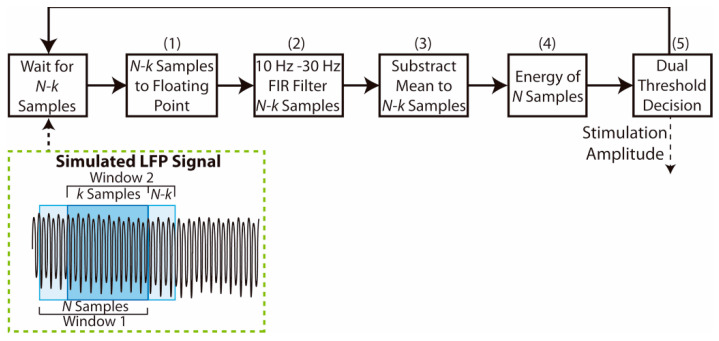
Dual threshold aDBS algorithm processing blocks implemented in the OS and moving overlapping window used. The algorithm is processed each *N-k* non-overlapping samples are obtained. Then, these samples are converted to floating point, filtered and the mean value is subtracted. The energy value of the full window (*N* samples) is obtained and evaluated against the dual threshold values to make a decision and change the stimulation amplitude.

**Figure 4 sensors-21-02349-f004:**
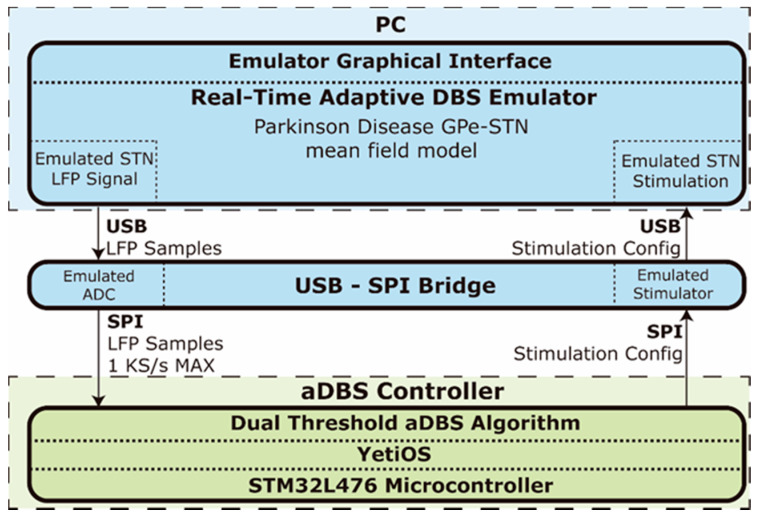
Emulation environment used in the experiments performed. On the bottom, the aDBS controller is the device under test whose power consumption is measured that runs the algorithm to dynamically change the stimulation amplitude. Its main hardware element is a STM32L476 microcontroller which runs YetiOS. The controller is coupled to a real-time emulator that provides the emulated subthalamic nucleus (STN) local field potential (LFP) samples through an USB to serial peripheral interface (SPI) bridge, which also allows the controller changing the stimulation amplitude of the emulator when the decision is made.

**Figure 5 sensors-21-02349-f005:**
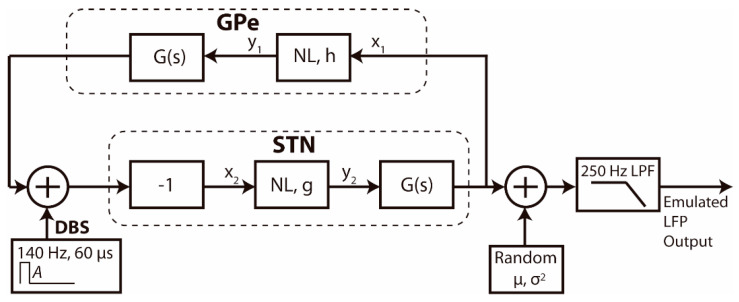
Schematic diagram of fourth order model of the GPe-STN loop including the pulse stimulator in the STN, a noise source and a simulated low pass filter (LPF) to simulate the physiological system and acquisition hardware. The stimulation is introduced in the GPe-STN loop [40] using a pulse generator with fixed 60 μs pulse width and 140 Hz frequency. The amplitude can be tuned by the aDBS algorithm which is in charge to process the emulated LFP output to make a decision.

**Figure 6 sensors-21-02349-f006:**
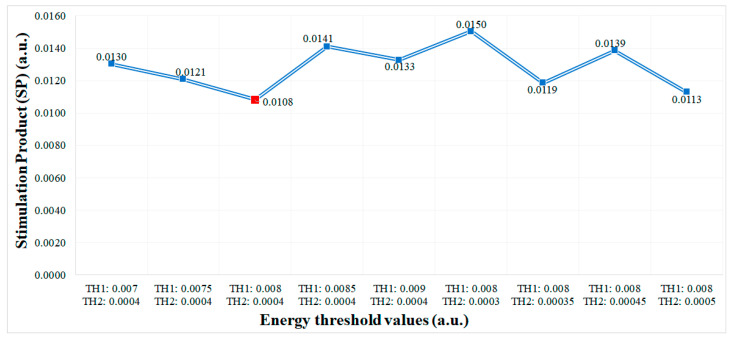
Performance in terms of stimulation product (*SP)* of the aDBS dual threshold algorithm for different energy threshold values. Performance is considered to be better for lower SP values. In this case, the threshold values TH1 = 0.008 a.u. and TH2 = 0.0004 a.u. provides the lowest SP (red point).

**Figure 7 sensors-21-02349-f007:**
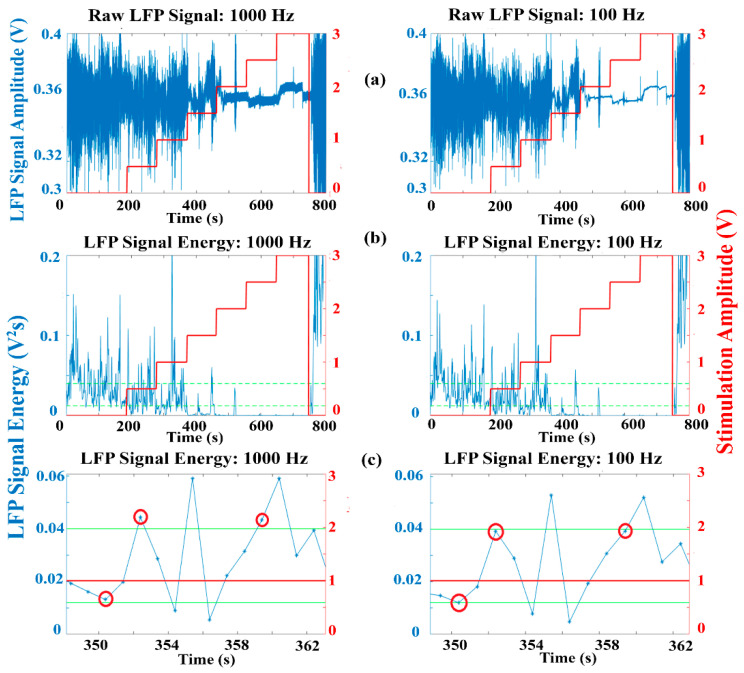
Local field potential (LFP) data recorded from a PD patient for increasing DBS amplitude (without aDBS) and downsampled to 1000 Hz and 100 Hz (**a**) LFP signal amplitude (blue) and DBS amplitude (red). (**b**) Calculated signal energy (blue) and DBS amplitude (red). (**c**) Detail of (**b**) indicating some points in which the dual threshold decision made is different for 1000 Hz and 100 Hz.

**Figure 8 sensors-21-02349-f008:**
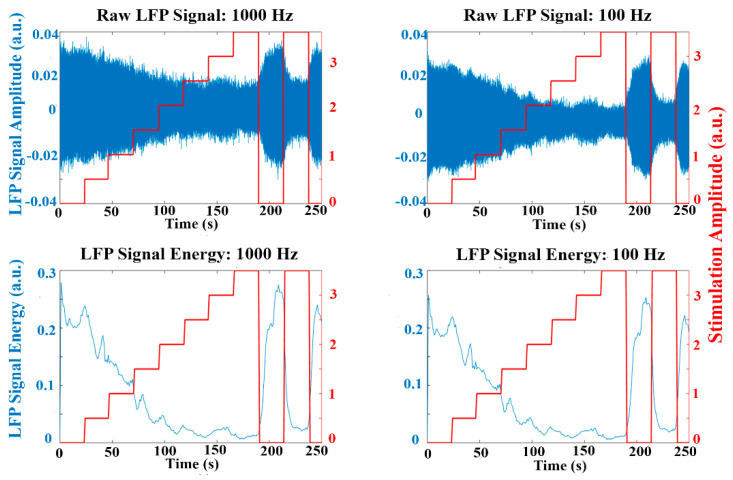
Raw LFP data (blue top), calculated signal energy (blue bottom) and stimulation signal (red) using the emulation environment for 1000 Hz and 100 Hz. Fixed increasing stimulation amplitude (without aDBS). When the stimulation amplitude increases, the LFP signal energy drops. A similar behaviour can be overserved compared to the data from a PD patient (Figure 7).

**Figure 9 sensors-21-02349-f009:**
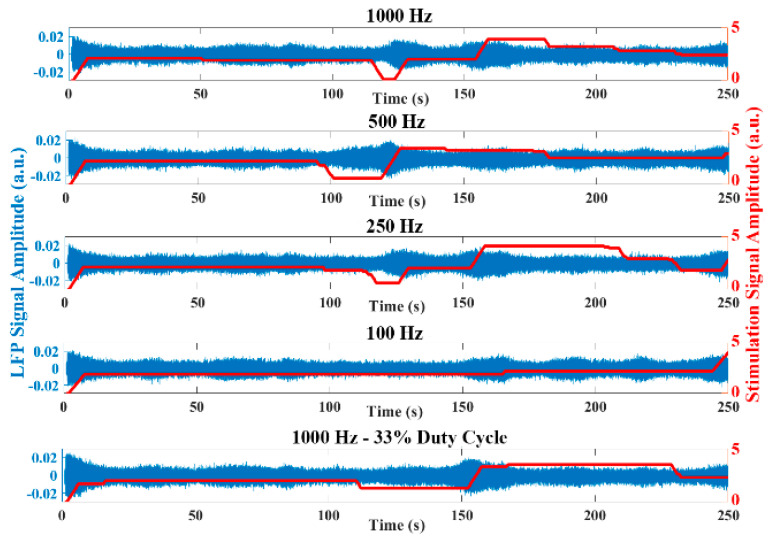
LFP signal amplitude (blue) and stimulation signal amplitude (red) generated by the aDBS algorithm for all the sample rates evaluated and the duty-cycling mode. Tunning the sample rate and the duty cycle leads to different decisions done by the aDBS algorithm to change the stimulation amplitude and to different LFP signals. Both have an influence in the performance in terms of *SP*.

**Figure 10 sensors-21-02349-f010:**
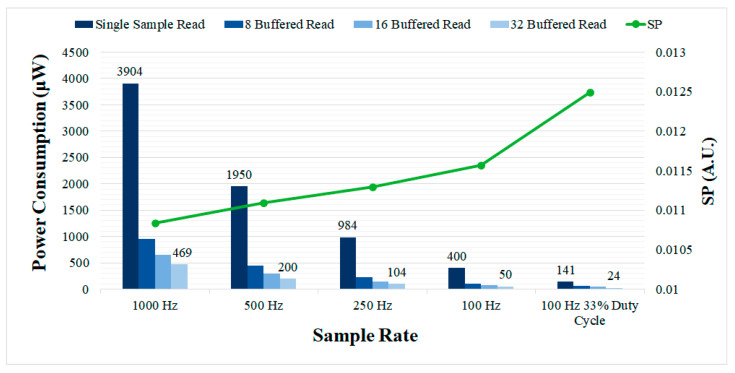
Power consumption and *SP* of the OS-based controller running the dual threshold aDBS algorithm for different sample rates and number of buffered read samples. The performance in terms of SP degrades for lower sample rates and when introducing duty cycling, while, in these cases, the power consumption is greatly reduced. Increasing the number of buffered samples improves the power consumption, while having no effect in the performance.

**Figure 11 sensors-21-02349-f011:**
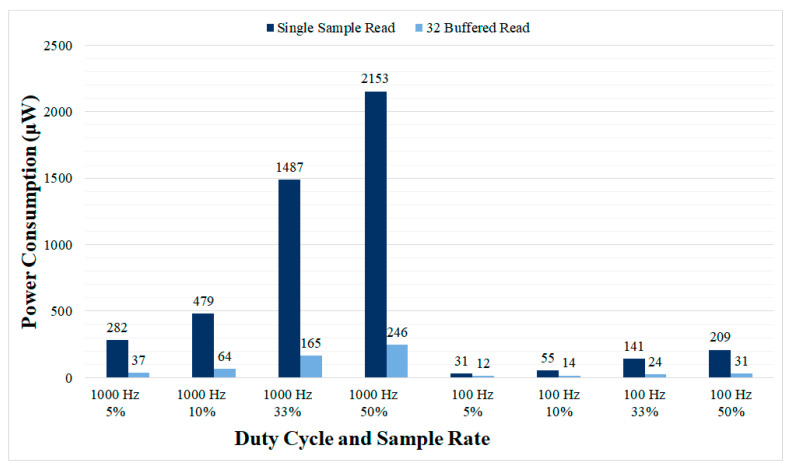
Power consumption of the OS-based controller running the dual threshold aDBS algorithm for different sample rates and duty cycle configurations. Lowering the sample rate, the duty cycle and using 32 samples buffered read results in introducing as low as 12 μW while providing full OS support running a dual threshold aDBS algorithm.

**Table 1 sensors-21-02349-t001:** Adaptive DBS algorithm parameters in the experiments performed.

Sample Rate (Hz)	Window Size Samples *N*	Non Overlap Samples *N-k*	FIR Filter Taps M	Decision Time (ms)
100	128	32	12	320
250	256	64	18	256
500	512	128	28	256
1000	1024	256	64	256

## Data Availability

Data available in a publicly accessible repository [42].

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
