# Peer review of "Methods for Lowering the Power Consumption of OS-Based Adaptive Deep Brain Stimulation Controllers"

_sensors, 2021, doi:10.3390/s21072349_

Round 1

Reviewer 1 Report

  1. General Comments
    1. It is not clear what the aim of the paper is.
      The abstract and introduction discuss the benefits and drawbacks of OS. It is stated that “...in this work we present an OS-based solution as the controller element of an aDBS device” (line 92) and “The analysis done in this work is intended to provide design guidelines for future aDBS devices which could substantially expand their capability and flexibility by using an OS as the core of their controller module” (line 114) but the OS seems not to be the creative work of this paper since there is not enough information to reproduce it. The important information is the power saving methods. The confusion is even bigger, when the title is taken into account, because it also implies more in depth discussion of the other elements of the system.
      The issue can be resolved by discussing OS in less detail by just stating their advantages and then focusing more on resolving the drawback. That can be accompanied with change in the title like “Methods for lowering the power consumption of adaptive brain stimulation implants”. That would also reduce the length of the introduction, which is on the longer side.
    2. The figures are hard to read and understand because their explanation is in the main text but not underneath the figures in the form of captions.
  2. Unclarities
    1. Stimulation Error Rate
      The SER is the main metric, however, its benefits as a metric are not clear. While it will definitely show statistically significant difference it is not sure that it relates to biologically/clinically significant one. At Fig. 7 c), 3 points at 100Hz are misclassified compared to at 1kHz, however, the points around them are classified correctly. Therefore, the lower frequency system would reach the same conclusion a bit later (about a second later). It is not convincing that a lag of 1 second is a significant measurement.
      In addition to that, using the 1kHz measurement as a gold standard is not advisable, I would propose using an already known and accepted algorithm.
    2. Application
      It is not clear if the application is meant to be used clinically in a device, which is already on the market, used for research or something else.
  3. Technical Comments
    1. In table 1, the FIR filters seem not to be equivalent between the experiments. A more detailed explanation why those values have been chosen will be beneficial.
  4. Additional Points of Interest Questions
    1. The switching from and into tickless mode is associated with power consumption (line 144). At what frequency the power needed to execute the command will be more than the power saved in the down time?
    2. The buffered read mode reduces the power consumption. It would be interesting to have tested how different number of samples in the bucket affect the performance and the power consumption (ex: 3, 5, 10, 50, 100 samples etc)

Reviewer 2 Report

• Thank you for the opportunity to review this manuscript.

• The paper describes an RTOS designed for adaptive DBS applications and describes the simulated sampling performance and power consumption of a device while employing sampling techniques aimed at reducing the power consumption overhead introduced by a more powerful device and an OS over a microcontroller and firmware.

• The manuscript is well written, and the experimentation easy to follow. However, I struggle with the aims of the paper and if it constitutes a significant advancement to the field and warrants a journal publication. It could be argued that the ease of programming is inconsequential for a medical device that is aimed towards changing a person’s life. Especially so if the ease of use comes at the expense of higher power consumption, and therefore more surgery, if not mitigated by a sampling technique that must reduce the response time. Please provide a more expanded motivation for why a more powerful framework, with the added power consumption, is required. 

• I bring this up since the authors mention that the DBS procedure is used to reduce essential tremor symptoms. This case is articulatory important to consider since the tremor is often an action tremor, that is introduced essentially by the person's intentional movement. If a patient is reaching for a coffee cup and gets tremor reduction much later, it will matter greatly for the patient and their ability to have a normal social interaction. So, the added processing delay that must generally be introduced by a read buffer matters in the application - or that is at least my thinking until the authors can demonstrate me wrong.

Reviewer 3 Report

The scope of this study is currently a significant research area. The manuscript introduces interesting technological findings. The paper is well written and organized. I only suggest two small improvements before publication.

- Line 34-35 (section ‘Introduction’):  What do you mean by the sentence «Deep Brain Stimulation (DBS) has become established as a surgical treatment?» Is DBS a "surgical treatment"?

- Section ‘Conclusion’: I invite authors to state that this YetiOS-based controller can also be used in other next generation bioelectronic intracorporeal implants, such as bioelectronics bone implants. I propose to cite the following papers in this scope:

Marco P. Soares dos Santos et al. (2021). "Towards an effective sensing technology to monitor micro-scale interface loosening of bioelectronic implants”, Scientific Reports, 11: 3449.

Reviewer 4 Report

This article proposes an OS-based solution to the power consumption problem faced within aDBS devices. Using different techniques, improvements in power management were achieved; thus demonstrating the potential for lower power consumption devices that are optimized at the OS-level boasting flexibility. Although clinical validity is still warranted, the study demonstrates its viability as proof of concept. The article is well written and adequately describes the methodological implementation. 

Author Response

The authors would like to thank the reviewers for their time and their valuable comments. We really appreciate their effort and interest.

Round 2

Reviewer 1 Report

no suggestions